# Gossip-based Actor-Learner Architectures for Deep Reinforcement Learning

**Mahmoud Assran**
Facebook AI Research &
Department of Electrical and Computer Engineering
McGill University
mahmoud.assran@mail.mcgill.ca

**Joshua Romoff**
Facebook AI Research &
Department of Computer Science
McGill University
joshua.romoff@mail.mcgill.ca

**Nicolas Ballas**
Facebook AI Research
ballasn@fb.com

**Joelle Pineau**
Facebook AI Research
jpineau@fb.com

**Michael Rabbat**
Facebook AI Research
mikerabbat@fb.com

## Abstract

Multi-simulator training has contributed to the recent success of Deep Reinforcement Learning by stabilizing learning and allowing for higher training throughputs. We propose Gossip-based Actor-Learner Architectures (GALA) where several actor-learners (such as A2C agents) are organized in a peer-to-peer communication topology, and exchange information through asynchronous gossip in order to take advantage of a large number of distributed simulators. We prove that GALA agents remain within an $\epsilon$-ball of one-another during training when using loosely coupled asynchronous communication. By reducing the amount of synchronization between agents, GALA is more computationally efficient and scalable compared to A2C, its fully-synchronous counterpart. GALA also outperforms A3C, being more robust and sample efficient. We show that we can run several loosely coupled GALA agents in parallel on a single GPU and achieve significantly higher hardware utilization and frame-rates than vanilla A2C at comparable power draws.

## 1 Introduction

Deep Reinforcement Learning (Deep RL) agents have reached superhuman performance in a few domains [Silver et al., 2016, 2018, Mnih et al., 2015, Vinyals et al., 2019], but this is typically at significant computational expense [Tian et al., 2019]. To both reduce running time and stabilize training, current approaches rely on distributed computation wherein data is sampled from many parallel simulators distributed over parallel devices [Espeholt et al., 2018, Mnih et al., 2016]. Despite the growing ubiquity of multi-simulator training, scaling Deep RL algorithms to a large number of simulators remains a challenging task.

On-policy approaches train a policy by using samples generated from that same policy, in which case data sampling (acting) is entangled with the training procedure (learning). To perform distributed training, these approaches usually introduce multiple learners with a *shared policy*, and multiple actors (each with its own simulator) associated to each learner. The shared policy can either be updated in a synchronous fashion (e.g., learners synchronize gradients before each optimization step [Stooke and Abbeel, 2018]), or in an asynchronous fashion [Mnih et al., 2016]. Both approaches have drawbacks: synchronous approaches suffer from straggler effects (bottlenecked by the slowest individual simulator), and therefore may not exhibit strong scaling efficiency; asynchronous methods are robust to stragglers, but prone to gradient staleness, and may become unstable with a large number of actors [Clemente et al., 2017].

Alternatively, off-policy approaches typically train a policy by sampling from a replay buffer of past transitions [Mnih et al., 2015]. Training off-policy allows for disentangling data-generation from learning, which can greatly increase computational efficiency when training with many parallel actors [Espeholt et al., 2018, Horgan et al., 2018, Kapturowski et al., 2019, Gruslys et al., 2018]. Generally, off-policy updates need to be handled with care as the sampled transitions may not conform to the current policy and consequently result in unstable training [Fujimoto et al., 2018].

We propose Gossip-based Actor-Learner Architectures (GALA), which aim to retain the robustness of synchronous on-policy approaches, while improving both their computational efficiency and scalability. GALA leverages multiple agents, where each agent is composed of one learner and possibly multiple actors/simulators. Unlike classical on-policy approaches, GALA does not require that each agent share the same policy, but rather it inherently enforces (through gossip) that each *agent's policy remain $\epsilon$-close to all others throughout training*. Relaxing this constraint allows us to reduce the synchronization needed between learners, thereby improving the algorithm's computational efficiency.

Instead of computing an exact average between all the learners after a local optimization step, gossip-based approaches compute an approximate average using loosely coupled and possibly asynchronous communication (see Nedić et al. [2018] and references therein). While this approximation implicitly injects some noise in the aggregate parameters, we prove that this is in fact a principled approach as the learners' policies stay within an $\epsilon$-ball of one-another (even with non-linear function approximation), the size of which is directly proportional to the spectral-radius of the agent communication topology and their learning rates.

As a practical algorithm, we propose GALA-A2C, an algorithm that combines gossip with A2C agents. We compare our approach on six Atari games [Machado et al., 2018] following Stooke and Abbeel [2018] with vanilla A2C, A3C and the IMPALA off-policy method [Dhariwal et al., 2017, Mnih et al., 2016, Espeholt et al., 2018]. Our main empirical findings are:

1. Following the theory, GALA-A2C is empirically stable. Moreover, we observe that GALA can be more stable than A2C when using a large number of simulators, suggesting that the noise introduced by gossiping can have a beneficial effect.

2. GALA-A2C has similar sample efficiency to A2C and greatly improves its computational efficiency and scalability.

3. GALA-A2C achieves significantly higher hardware utilization and frame-rates than vanilla A2C at comparable power draws, when using a GPU.

4. GALA-A2C is competitive in term of performance relative to A3C and IMPALA.

Perhaps most remarkably, our empirical findings for GALA-A2C are obtained by simply using the default hyper-parameters from A2C. Our implementation of GALA-A2C is publicly available at https://github.com/facebookresearch/gala.

## 2  Technical Background

**Reinforcement Learning.**   We consider the standard Reinforcement Learning setting [Sutton and Barto, 1998], where the agent's objective is to maximize the expected value from each state $V(s) = \mathbb{E}\left[\sum_{i=0}^{\infty} \gamma^i r_{t+i} | s_t = s\right]$, $\gamma$ is the discount factor which controls the bias towards nearby rewards. To maximize this quantity, the agent chooses at each discrete time step $t$ an action $a_t$ in the current state $s_t$ based on its policy $\pi(a_t|s_t)$ and transitions to the next state $s_{t+1}$ receiving reward $r_t$ based on the environment dynamics.

Temporal difference (TD) learning [Sutton, 1984] aims at learning an approximation of the expected return parameterized by $\theta$, i.e., the value function $V(s; \theta)$, by iteratively updating its parameters via gradient descent:

$$\nabla_\theta \left(G_t^N - V(s_t; \theta)\right)^2 \tag{1}$$

where $G_t^N = \sum_{i=0}^{N-1} \gamma^i r_{t+i} + \gamma^N V(s_{t+n}; \theta_t)$ is the $N$-step return. Actor-critic methods [Sutton et al., 2000, Mnih et al., 2016] simultaneously learn both a parameterized policy $\pi(a_t|s_t; \omega)$ with parameters $\omega$ and a critic $V(s_t; \theta)$. They do so by training a value function via the TD error defined

in (1) and then proceed to optimize the policy using the policy gradient (PG) with the value function as a baseline:

$$\nabla_\omega\left(-\log\pi(a_t|s_t;\omega)A_t\right),\tag{2}$$

where $A_t = G_t^N - V(s_t;\theta_t)$ is the advantage function, which represents the relative value the current action has over the average. In order to both speed up training time and decorrelate observations, Mnih et al. [2016] collect samples and perform updates with several asynchronous actor-learners. Specifically, each worker $i \in \{1,2,..,W\}$, where $W$ is the number of parallel workers, collects samples according to its current version of the policy weights $\omega_i$, and computes updates via the standard actor-critic gradient defined in (2), with an additional entropy penalty term that prevents premature convergence to deterministic policies:

$$\nabla_{\omega_i}\left(-\log\pi(a_t|s_t;\omega_i)A_t - \eta\sum_a \pi(a|s_t;\omega_i)\log\pi(a|s_t;\omega_i)\right).\tag{3}$$

The workers then perform HOGWILD! [Recht et al., 2011] style updates (asynchronous writes) to a shared set of master weights before synchronizing their weights with the master's. More recently, Dhariwal et al. [2017] removed the asynchrony from A3C, referred to as A2C, by instead synchronously collecting transitions in parallel environments $i \in \{1,2,..,W\}$ and then performing a large batched update:

$$\nabla_\omega\left[\frac{1}{W}\sum_{i=1}^{W}\left(-\log\pi(a_t^i|s_t^i;\omega)A_t^i - \eta\sum_a \pi(a|s_t^i;\omega)\log\pi(a|s_t^i;\omega)\right)\right].\tag{4}$$

**Gossip algorithms.**  Gossip algorithms are used to solve the distributed averaging problem. Suppose there are $n$ agents connected in a peer-to-peer graph topology, each with parameter vector $x_i^{(0)} \in \mathbb{R}^d$. Let $\boldsymbol{X}^{(0)} \in \mathbb{R}^{n\times d}$ denote the row-wise concatenation of these vectors. The objective is to iteratively compute the average vector $\frac{1}{n}\sum_{i=1}^n \boldsymbol{x}_i^{(0)}$ across all agents. Typical gossip iterations have the form $\boldsymbol{X}^{(k+1)} = \boldsymbol{P}^{(k)}\boldsymbol{X}^{(k)}$, where $\boldsymbol{P}^{(k)} \in \mathbb{R}^{n\times n}$ is referred to as the mixing matrix and defines the communication topology. This corresponds to the update $\boldsymbol{x}_i^{(k+1)} = \sum_{j=1}^n p_{i,j}^{(k)}\boldsymbol{x}_j^{(k)}$ for an agent $v_i$. At an iteration $k$, an agent $v_i$ only needs to receive messages from other agents $v_j$ for which $p_{i,j}^{(k)} \neq 0$, so sparser matrices $\boldsymbol{P}^{(k)}$ correspond to less communication and less synchronization between agents.

The mixing matrices $\boldsymbol{P}^{(k)}$ are designed to be row stochastic (each entry is greater than or equal to zero, and each row sums to 1) so that $\lim_{K\to\infty}\prod_{k=0}^K \boldsymbol{P}^{(k)} = \mathbf{1}\boldsymbol{\pi}^\top$, where $\boldsymbol{\pi}$ is the ergodic limit of the Markov chain defined by $\boldsymbol{P}^{(k)}$ and $\mathbf{1}$ is a vector with all entries equal to 1 [Seneta, 1981].[1] Consequently, the gossip iterations converge to a limit $\boldsymbol{X}^{(\infty)} = \mathbf{1}(\boldsymbol{\pi}^\top\boldsymbol{X}^{(0)})$; meaning the value at an agent $i$ converges to $\boldsymbol{x}_i^{(\infty)} = \sum_{j=1}^n \pi_j\boldsymbol{x}_j^{(0)}$. In particular, if the matrices $\boldsymbol{P}^{(k)}$ are symmetric and doubly-stochastic (each row and each column must sum to 1), we obtain an algorithm such that $\pi_j = 1/n$ for all $j$, and therefore $\boldsymbol{x}_i^{(\infty)} = 1/n\sum_{j=1}^n \boldsymbol{x}_j^{(0)}$ converges to the average of the agents' initial vectors.

For the particular case of GALA, we only require the matrices $\boldsymbol{P}^{(k)}$ to be row stochastic in order to show the $\epsilon$-ball guarantees.

## 3  Gossip-based Actor-Learner Architectures

We consider the distributed RL setting where $n$ agents (each composed of a single learner and several actors) collaborate to maximize the expected return $V(s)$. Each agent $v_i$ has a parameterized policy network $\pi(a_t|s_t;\omega_i)$ and value function $V(s_t;\theta_i)$. Let $x_i = (\omega_i,\theta_i)$ denote agent $v_i$'s complete set of trainable parameters. We consider the specific case where each $v_i$ corresponds to a single A2C agent, and the agents are configured in a directed and peer-to-peer communication topology defined by the mixing matrix $\boldsymbol{P} \in \mathbb{R}^{n\times n}$.

In order to maximize the expected reward, each GALA-A2C agent alternates between one local policy-gradient and TD update, and one iteration of asynchronous gossip with its peers. Pseudocode is

**Algorithm 1** Gossip-based Actor-Learner Architectures for agent $v_i$ using A2C

---

**Require:** Initialize trainable policy and critic parameters $x_i = (\omega_i, \theta_i)$.
 1: **for** $t = 0, 1, 2, \dots$ **do**
 2:     Take $N$ actions $\{a_t\}$ according to $\pi_{\omega_i}$ and store transitions $\{(s_t, a_t.r_t, s_{t+1})\}$
 3:     Compute returns $G_t^N = \sum_{i=0}^{N-1} \gamma^i r_{t+i} + \gamma^N V(s_{t+n}; \theta_i)$ and advantages $A_t = G_t^N - V(s_t; \theta_i)$
 4:     Perform A2C optimization step on $x_i$ using TD in (1) and batched policy-gradient in (4)
 5:     Broadcast (non-blocking) new parameters $x_i$ to all out-peers in $\mathcal{N}_i^{\text{out}}$
 6:     **if** Receive buffer contains a message $m_j$ from each in-peer $v_j$ in $\mathcal{N}_i^{\text{in}}$ **then**
 7:         $x_i \leftarrow \frac{1}{1+|\mathcal{N}_i^{\text{in}}|}(x_i + \sum_j m_j)^1$         ▷ Average parameters with messages
 8:     **end if**
 9: **end for**

---

$^1$ We set the non-zero mixing weights for agent $v_i$ to $p_{i,j} = \frac{1}{1+|\mathcal{N}_i^{\text{in}}|}$.

---

provided in Algorithm 1, where $\mathcal{N}_i^{\text{in}} := \{v_j \mid p_{i,j} > 0\}$ denotes the set of agents that send messages to agent $v_i$ (in-peers), and $\mathcal{N}_i^{\text{out}} := \{v_j \mid p_{j,i} > 0\}$ the set of agents that $v_i$ sends messages to (out-peers). During the gossip phase, agents broadcast their parameters to their out-peers, asynchronously (i.e., don't wait for messages to reach their destination), and update their own parameters via a convex combination of all received messages. Agents broadcast new messages when old transmissions are completed and aggregate all received messages once they have received a message from each in-peer.

Note that the GALA agents use non-blocking communication, and therefore operate asynchronously. Local iteration counters may be out-of-sync, and physical message delays may result in agents incorporating outdated messages from their peers. One can algorithmically enforce an upper bound on the message staleness by having the agent block and wait for communication to complete if more than $\tau \geq 0$ local iterations have passed since the agent last received a message from its in-peers.

**Theoretical $\epsilon$-ball guarantees:**   Next we provide the $\epsilon$-ball theoretical guarantees for the asynchronous GALA agents, proofs of which can be found in Appendix B. Let $k \in \mathbb{N}$ denote the global iteration counter, which increments whenever any agent (or subset of agents) completes an iteration of the loop defined in Algorithm 1. We define $x_i^{(k)} \in \mathbb{R}^d$ as the value of agent $v_i$'s trainable parameters at iteration $k$, and $\boldsymbol{X}^{(k)} \in \mathbb{R}^{n \times d}$ as the row-concatenation of these parameters.

For our theoretical guarantees we let the communication topologies be directed and time-varying graphs, and we do not make any assumptions about the base GALA learners. In particular, let the mapping $\mathcal{T}_i : x_i^{(k)} \in \mathbb{R}^d \mapsto x_i^{(k)} - \alpha g_i^{(k)} \in \mathbb{R}^d$ characterize agent $v_i$'s local training dynamics (i.e., agent $v_i$ optimizes its parameters by computing $x_i^{(k)} \leftarrow \mathcal{T}_i(x_i^{(k)})$), where $\alpha > 0$ is a reference learning rate, and $g_i^{(k)} \in \mathbb{R}^d$ can be any update vector. Lastly, let $\boldsymbol{G}^{(k)} \in \mathbb{R}^{n \times d}$ denote the row-concatenation of these update vectors.

**Proposition 1.** *For all $k \geq 0$, it holds that*

$$\left\| \boldsymbol{X}^{(k+1)} - \overline{\boldsymbol{X}}^{(k+1)} \right\| \leq \alpha \sum_{s=0}^{k} \beta^{k+1-s} \left\| \boldsymbol{G}^{(s)} \right\|,$$

*where $\overline{\boldsymbol{X}}^{(k+1)} := \frac{1_n 1_n^T}{n} \boldsymbol{X}^{(k+1)}$ denotes the average of the learners' parameters at iteration $k + 1$, and $\beta \in [0, 1]$ is related to the joint spectral radius of the graph sequence defining the communication topology at each iteration.*

Proposition 1 shows that the distance of a learners' parameters from consensus is bounded at each iteration. However, without additional assumptions on the communication topology, the constant $\beta$ may equal 1, and the bound in Proposition 1 can be trivial. In the following proposition, we make sufficient assumptions with respect to the graph sequence that ensure $\beta < 1$.

**Proposition 2.** *Suppose there exists a finite integer $B \geq 0$ such that the (potentially time-varying) graph sequence is $B$-strongly connected, and suppose that the upper bound $\tau$ on the message delays in Algorithm 1 is finite. If learners run Algorithm 1 from iteration 0 to $k + 1$, where $k \geq \tau + B$, then it holds that*

$$\left\| \boldsymbol{X}^{(k+1)} - \overline{\boldsymbol{X}}^{(k+1)} \right\| \leq \frac{\alpha \tilde{\beta} L}{1 - \beta},$$

*where $\beta < 1$ is related to the joint spectral radius of the graph sequence, $\alpha$ is the reference learning rate, $\tilde{\beta} := \beta^{-\frac{\tau+B}{\tau+B+1}}$, and $L := \sup_{s=1,2,...} \left\| G^{(s)} \right\|$ denotes an upper bound on the magnitude of the local optimization updates during training.*

Proposition 2 states that the agents' parameters are guaranteed to reside within an $\epsilon$-ball of their average at all iterations $k \geq \tau + B$. The size of this ball is proportional to the reference learning-rate, the spectral radius of the graph topology, and the upper bound on the magnitude of the local gradient updates. One may also be able to control the constant $L$ in practice since Deep RL agents are typically trained with some form of gradient clipping.

## 4   Related work

Several recent works have approached scaling up RL by using parallel environments. Mnih et al. [2016] used parallel asynchronous agents to perform HOGWILD! [Recht et al., 2011] style updates to a shared set of parameters. Dhariwal et al. [2017] proposed A2C, which maintains the parallel data collection, but performs updates synchronously, and found this to be more stable empirically. While A3C was originally designed as a purely CPU-based method, Babaeizadeh et al. [2017] proposed GA3C, a GPU implementation of the algorithm. Stooke and Abbeel [2018] also scaled up various RL algorithms by using significantly larger batch sizes and distributing computation onto several GPUs. Differently from those works, we propose the use of *Gossip Algorithms* to aggregate information between different agents and thus simulators. Nair et al. [2015], Horgan et al. [2018], Espeholt et al. [2018], Kapturowski et al. [2019], Gruslys et al. [2018] use parallel environments as well, but disentangle the data collection (actors) from the network updates (learners). This provides several computational benefits, including better hardware utilization and reduced straggler effects. By disentangling acting from learning these algorithms must use off-policy methods to handle learning from data that is not directly generated from the current policy (e.g., slightly older policies).

Gossip-based approaches have been extensively studied in the control-systems literature as a way to aggregate information for distributed optimization algorithms [Nedić et al., 2018]. In particular, recent works have proposed to combine gossip algorithms with stochastic gradient descent in order to train Deep Neural Networks [Lian et al., 2018, 2017, Assran et al., 2019], but unlike our work, focus only on the supervised classification paradigm.

## 5   Experiments

We evaluate GALA for training Deep RL agents on Atari-2600 games [Machado et al., 2018]. We focus on the same six games studied in Stooke and Abbeel [2018]. Unless otherwise-stated, all learning curves show averages over 10 random seeds with $95\%$ confidence intervals shaded in. We follow the reproducibility checklist [Pineau, 2018], see Appendix A for details.

We compare A2C [Dhariwal et al., 2017], A3C [Mnih et al., 2016], IMPALA [Espeholt et al., 2018], and GALA-A2C. All methods are implemented in PyTorch [Paszke et al., 2017]. While A3C was originally proposed with CPU-based agents with 1-simulator per agent, Stooke and Abbeel [2018] propose a large-batch variant in which each agent manages 16-simulators and performs batched inference on a GPU. We found this large-batch variant to be more stable and computationally efficient (cf. Appendix C.1). We use the Stooke and Abbeel [2018] variant of A3C to provide a more competitive baseline. We parallelize A2C training via the canonical approach outlined in Stooke and Abbeel [2018], whereby individual A2C agents (running on potentially different devices), all average their gradients together before each update using the ALLREDUCE primitive.[2] For A2C and A3C we use the hyper-parameters suggested in Stooke and Abbeel [2018]. For IMPALA we use the hyper-parameters suggested in Espeholt et al. [2018]. For GALA-A2C we use the same hyper-parameters as the original (non-gossip-based) method. All GALA agents are configured in a directed ring graph. All implementation details are described in Appendix C. For the IMPALA baseline, we use a prerelease of TorchBeast [Küttler et al., 2019] available at https://github.com/facebookresearch/torchbeast.

Table 1: Across all training seeds we select the best final policy produced by each method at the end of training and evaluate it over 10 evaluation episodes (up to 30 no-ops at the start of the episode). Evaluation actions generated from $\arg\max_a \pi(a|s)$. The table depicts the mean and standard error across these 10 evaluation episodes.

| | Steps | BeamRider | Breakout | Pong | Qbert | Seaquest | SpaceInvaders |
|---|---|---|---|---|---|---|---|
| IMPALA[1] | 50M | 8220 | 641 | **21** | 18902 | 1717 | 1727 |
| IMPALA | 40M | 7118 $_{\pm 2536}$ | 127 $_{\pm 65}$ | **21** $_{\pm 0}$ | 7878 $_{\pm 2573}$ | 462 $_{\pm 2}$ | **4071** $_{\pm 393}$ |
| A3C | 40M | 5674 $_{\pm 752}$ | 414 $_{\pm 56}$ | **21** $_{\pm 0}$ | 14923 $_{\pm 460}$ | 1840 $_{\pm 0}$ | 2232 $_{\pm 302}$ |
| A2C | 25M | 8755 $_{\pm 811}$ | 419 $_{\pm 3}$ | **21** $_{\pm 0}$ | 16805 $_{\pm 172}$ | 1850 $_{\pm 5}$ | 2846 $_{\pm 22}$ |
| A2C | 40M | **9829** $_{\pm 1355}$ | 495 $_{\pm 57}$ | **21** $_{\pm 0}$ | 19928 $_{\pm 99}$ | **1894** $_{\pm 6}$ | 3021 $_{\pm 36}$ |
| GALA-A2C | 25M | **9500** $_{\pm 1020}$ | **690** $_{\pm 72}$ | **21** $_{\pm 0}$ | 18810 $_{\pm 37}$ | 1874 $_{\pm 4}$ | 2726 $_{\pm 189}$ |
| GALA-A2C | 40M | **10188** $_{\pm 1316}$ | **690** $_{\pm 72}$ | **21** $_{\pm 0}$ | **20150** $_{\pm 28}$ | **1892** $_{\pm 6}$ | 3074 $_{\pm 69}$ |

[1] Espeholt et al. [2018] results using shallow network (identical to the network used in our experiments).

**Convergence and stability:** We begin by empirically studying the convergence and stability properties of A2C and GALA-A2C. Figure 1a depicts the percentage of successful runs (out of 10 trials) of standard policy-gradient A2C when we sweep the number of simulators across six different games. We define a run as successful if it achieves better than $50\%$ of nominal 16-simulator A2C scores. When using A2C, we observe an identical trend across all games in which the number of successful runs decreases significantly as we increase the number of simulators. Note that the A2C batch size is proportional to the number of simulators, and when increasing the number of simulators we adjust the learning rate following the recommendation in Stooke and Abbeel [2018].

Figure 1a also depicts the percentage of successful runs when A2C agents communicate their parameters using gossip algorithms (GALA-A2C). In *every* simulator sweep across the six games (600 runs), the gossip-based architecture increases or maintains the percentage of successful runs relative to vanilla A2C, when using identical hyper-parameters. We hypothesize that exercising slightly different policies at each learner using gossip-algorithms can provide enough decorrelation in gradients to improve learning stability. We revisit this point later on (cf. Figure 3b). We note that Stooke and Abbeel [2018] find that stepping through a random number of uniform random actions at the start of training can partially mitigate this stability issue. We did not use this random start action mitigation in the reported experiments.

While Figure 1a shows that GALA can be used to stabilize multi-simulator A2C and increase the number of successfull runs, it does not directly say anything about the final performance of the learned models. Figures 1b and 1c show the rewards plotted against the number of environment steps when training with 64 simulators. Using gossip-based architectures stabilizes and maintains the peak performance and sample efficiency of A2C across all six games (Figure 1b), and also increases the number of convergent runs (Figure 1c).

Figures 1d and 1e compare the wall-clock time convergence of GALA-A2C to vanilla A2C. Not only is GALA-A2C more stable than A2C, but it also runs at a higher frame-rate by mitigating straggler effects. In particular, since GALA-A2C learners do not need to synchronize their gradients, each learner is free to run at its own rate without being hampered by variance in peer stepping times.

**Comparison with distributed Deep RL approaches:** Figure 1 also compares GALA-A2C to state-of-the-art methods like IMPALA and A3C.[3] In each game, the GALA-A2C learners exhibited good sample efficiency and computational efficiency, and achieved highly competitive final game scores. Next we evaluate the final policies produced by each method at the end of training. After training across 10 different seeds, we are left with 10 distinct policies per method. We select the best final policy and evaluate it over 10 evaluation episodes, with actions generated from $\arg\max_a \pi(a|s)$. In almost every single game, the GALA-A2C learners achieved the highest evaluation scores of any method. Notably, the GALA-A2C learners that were trained for 25M steps achieved (and in most cases surpassed) the scores for IMPALA learners trained for 50M steps [Espeholt et al., 2018].

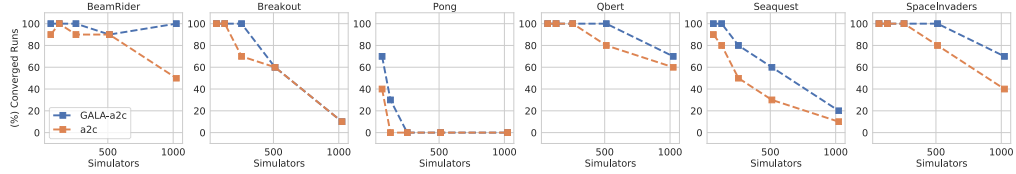

(a) Simulator sweep: Percentage of convergent runs out of 10 trials.

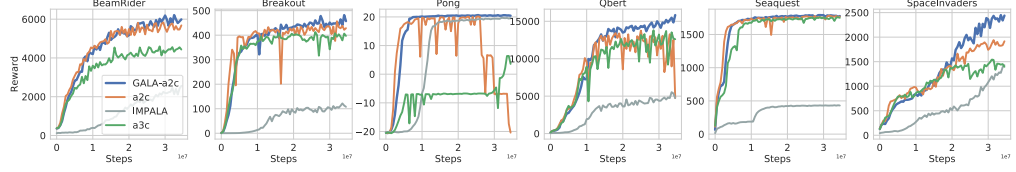

(b) Sample complexity: Best 3 runs for each method.

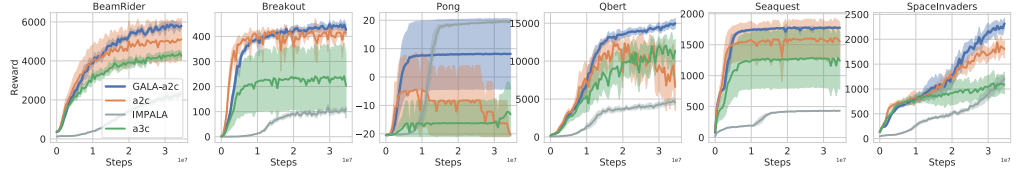

(c) Sample complexity: Average across 10 runs.

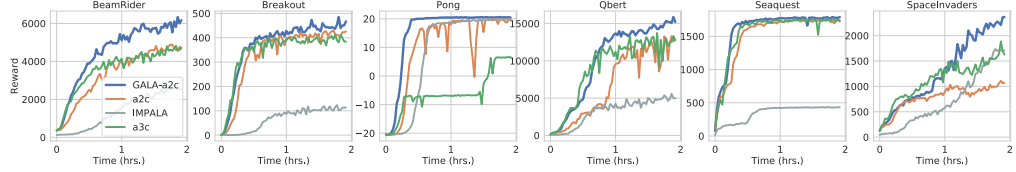

(d) Computational complexity: Best 3 runs for each method.

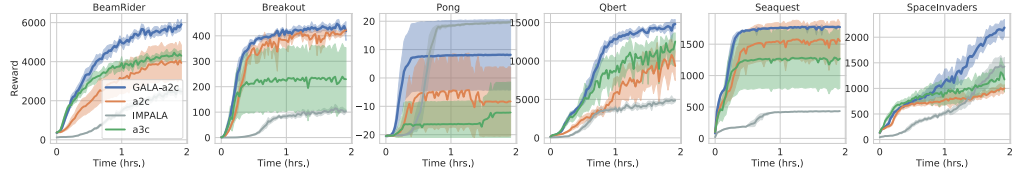

(e) Computational complexity: Average across 10 runs.

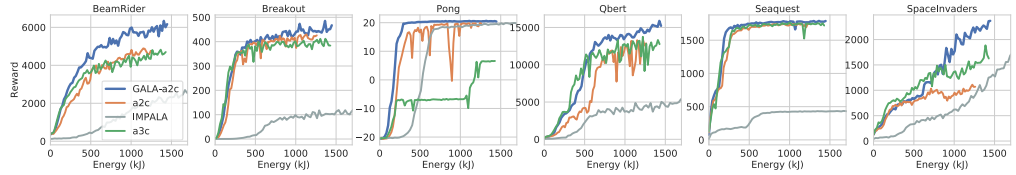

(f) Energy efficiency: Best 3 runs for each method.

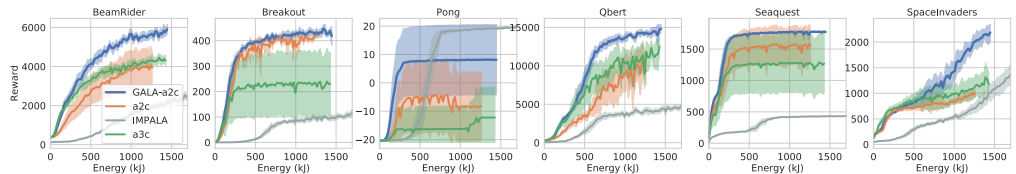

(g) Energy efficiency: Average across 10 runs.

Figure 1: (a) GALA increases or maintains the percentage of convergent runs relative to A2C. (b)-(c) GALA maintains the best performance of A2C while being more robust. (d)-(e) GALA achieves competitive scores in each game and in the shortest amount of time. (f)-(g) GALA achieves competitive game scores while being energy efficient.

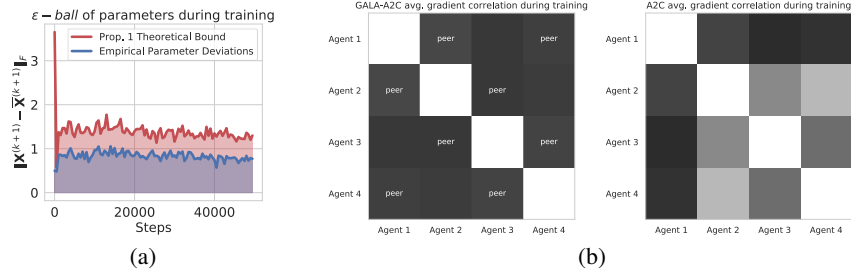

(a)                                                                          (b)

Figure 2: (a) The radius of the $\epsilon$-ball within which the agents' parameters reside during training. The theoretical upper bound in Proposition 1 is explicitly calculated and compared to the true empirical quantity. The bound in Proposition 1 is remarkably tight. (b) Average correlation between agents' gradients during training (darker colors depict low correlation and lighter colors depict higher correlations). Neighbours in the GALA-A2C topology are annotated with the label "peer." The GALA-A2C heatmap is generally much darker than the A2C heatmap, indicating that GALA-A2C agents produce more diverse gradients with significantly less correlation.

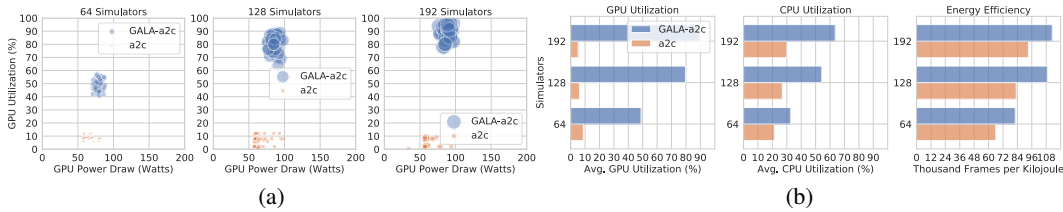

(a)                                                                          (b)

Figure 3: Comparing GALA-A2C hardware utilization to that of A2C when using one NVIDIA V100 GPU and 48 Intel CPUs. (a) Samples of instantaneous GPU utilization and power draw plotted against each other. Bubble sizes indicate frame-rates obtained by the corresponding algorithms; larger bubbles depict higher frame-rates. GALA-A2C achieves higher hardware utilization than A2C at comparable power draws. This translates to much higher frame-rates and increased energy efficiency. (b) Hardware utilization/energy efficiency vs. number of simulators. GALA-A2C benefits from increased parallelism and achieves a 10-fold improvement in GPU utilization over A2C.

**Effects of gossip:**  To better understand the stabilizing effects of GALA, we evaluate the diversity in learner policies during training. Figure 2a shows the distance of the agents' parameters from consensus throughout training. The theoretical upper bound in Proposition 1 is also explicitly calculated and plotted in Figure 2a. As expected, the learner policies remain within an $\epsilon$-ball of one-another in weight-space, and this size of this ball is remarkably well predicted by Proposition 1.

Next, we measure the diversity in the agents' gradients. We hypothesize that the $\epsilon$-diversity in the policies predicted by Proposition 1, and empirically observed in Figure 2a, may lead to less correlation in the agents' gradients. The categorical heatmap in Figure 2b shows the pair-wise cosine-similarity between agents' gradients throughout training, computed after every $500$ local environment steps, and averaged over the first 10M training steps. Dark colors depict low correlations and light colors depict high correlations. We observe that GALA-A2C agents exhibited less gradient correlations than A2C agents. Interestingly, we also observe that GALA-A2C agents' gradients are more correlated with those of peers that they explicitly communicate with (graph neighbours), and less correlated with those of agents that they do not explicitly communicate with.

**Computational performance:**  Figure 3 showcases the hardware utilization and energy efficiency of GALA-A2C compared to A2C as we increase the number of simulators. Specifically, Figure 3a shows that GALA-A2C achieves significantly higher hardware utilization than vanilla A2C at comparable power draws. This translates to much higher frame-rates and increased energy efficiency. Figure 3b shows that GALA-A2C is also better able to leverage increased parallelism and achieves a 10-fold improvement in GPU utilization over vanilla A2C. Once again, the improved hardware utilization and frame-rates translate to increased energy efficiency. In particular, GALA-A2C steps

through roughly 20 thousand more frames per Kilojoule than vanilla A2C. Figures 1f and 1g compare game scores as a function of energy utilization in Kilojoules. GALA-A2C is distinctly more energy efficient than the other methods, achieving higher game scores with less energy utilization.

## 6 Conclusion

We propose Gossip-based Actor-Learner Architectures (GALA) for accelerating Deep Reinforcement Learning by leveraging parallel actor-learners that exchange information through asynchronous gossip. We prove that the GALA agents' policies are guaranteed to remain within an $\epsilon$-ball during training, and verify this empirically as well. We evaluated our approach on six Atari games, and find that GALA-A2C improves the computational efficiency of A2C, while also providing extra stability and robustness by decorrelating gradients. GALA-A2C also achieves significantly higher hardware utilization than vanilla A2C at comparable power draws, and is competitive with state-of-the-art methods like A3C and IMPALA.

**Acknowledgments**

We would like to thank the authors of TorchBeast for providing their pytorch implementation of IMPALA.

## Footnotes

[1] Assuming that information from every agent eventually reaches all other agents

[2]This is mathematically equivalent to a single A2C agent with multiple simulators (e.g., $n$ agents, with $b$ simulators each, are equivalent to a single agent with $nb$ simulators).

[3]We report results for both the TorchBeast implementation of IMPALA, and from Table $C$.1 from Espeholt et al. [2018]

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
