[Supplementary Material · gala-supp.pdf]

# Appendices

## A  Reproducibility Checklist

We follow the reproducibility checklist Pineau [2018]: For all algorithms presented, check if you include:

- **A clear description of the algorithm.** See Algorithm 1
- **An analysis of the complexity (time, space, sample size) of the algorithm.** See Figures 1 and 3 for an analysis of sample efficiency, wall-clock time, and energy efficiency.
- **A link to a downloadable source code, including all dependencies.** See the attached zip file.

For any theoretical claim, check if you include:

- **A statement of the result.** See Propositions 1 and 2 in the main text.
- **A clear explanation of any assumptions.** See Appendix B for full details.
- **A complete proof of the claim.** See Appendix B for full details.

For all figures and tables that present empirical results, check if you include:

- **A complete description of the data collection process, including sample size.** We used the Arcade Learning Environment [Machado et al., 2018], specifically we used the gym package - see: github.com/openai/gym
- **A link to downloadable version of the dataset or simulation environment.** See: github.com/openai/gym
- **An explanation of how samples were allocated for training / validation / testing.** We didn't use explicit training / validation/ testing splits - but ran each algorithm with 10 different random seeds.
- **An explanation of any data that were excluded.** We only used 6 atari games due to time constraints - the same 6 games that were used in Stooke and Abbeel [2018].
- **The range of hyper-parameters considered, method to select the best hyper-parameter configuration, and specification of all hyper-parameters used to generate results.** We used standard hyper-parameters from Dhariwal et al. [2017], Stooke and Abbeel [2018], Espeholt et al. [2018].
- **The exact number of evaluation runs.** We used 10 seeds for the Atari experiments.
- **A description of how experiments were run.** See Appendix C for full details.
- **A clear definition of the specific measure or statistics used to report results.** $95\%$ confidence intervals are used in all plots / tables unless otherwise stated.
- **Clearly defined error bars.** $95\%$ confidence intervals are used in all plots / tables unless otherwise stated.
- **A description of results with central tendency (e.g. mean) and variation (e.g. stddev).** $95\%$ confidence intervals are used in all plots / tables unless otherwise stated.
- **A description of the computing infrastructure used.** See Appendix C for full details.

## B  Proofs

**Setting and Notation**  Before presenting the theoretical guarantees, we define some notation. Suppose we have $n$ learners (e.g., actor-critic agents) configured in a peer-to-peer communication topology represented by a directed and potentially time-varying graph (the non-zero entries in the mixing matrix $\boldsymbol{P}^{(k)}$ define the communication topology at each iteration $k$).

Learners constitute vertices in the graph, denoted by $v_i$ for all $i \in [n]$ , and edges constitute directed communication links. Let $\mathcal{N}_i^{\text{out}}$ denote agent $v_i$'s *out-peers*, the set of agents that $v_i$ can send messages

to, and let $\mathcal{N}_i^{\text{in}}$ denote agent $v_i$'s *in-peers*, the set of agents that can send messages to $v_i$. If the graph is time-varying, these sets are annotated with time indices. Let $x_i \in \mathbb{R}^d$ denote the agent $v_i$'s complete set of trainable parameters, and let the training function $\mathcal{T}_i : \mathbb{R}^d \mapsto \mathbb{R}^d$ define agent $v_i$'s training dynamics (i.e., agent $v_i$ optimizes its parameters by iteratively computing $x_i \leftarrow \mathcal{T}_i(x_i)$).

For each agent $v_i$ we define send- and receive-buffers, $\mathcal{B}_i$ and $\mathcal{R}_i$ respectively, which are used by the underlying communication system (standard in the gossip literature [Tsitsiklis et al., 1986]). When an agent wishes to broadcast a message to its out-peers, it simply copies the message into its broadcast buffer. Similarly, when agent receives a message, it is automatically copied into the receive buffer. For convenience, we assume that each learner $v_i$ can hold at most one message from in-peer $v_j$ in its receive buffer, $\mathcal{R}_i$ at any time $k$; i.e., a newly received message from agent $v_j$ overwrites the older one in the receive buffer.

Let $k \in \mathbb{N}$ denote the global iteration counter. That is, $k$ increments whenever any agent (or subset of agents) completes one loop in Algorithm 1. Consequently, at each global iteration $k$, there is a set of agents $\mathcal{I}$ that are activated, and within this set there is a (possibly non-empty) subset of agents $\mathcal{C} \subseteq \mathcal{I}$ that gossip in the same iteration. If a message from agent $v_j$ is received by agent $v_i$ at time $k$, let $\tau_{j,i}^{(k)}$ denote the time at which this message was sent. Let $\tau \geq \tau_{j,i}^{(k)}$ for all $i, j \in [n]$ and $k > 0$ denote an upper bound on the message delays. For analysis purposes, messages are sent with an effective delay such that they arrive right when the agent is ready to process the messages. That is, a message that is sent by agent $v_j$ at iteration $k'$ and processed by agent $v_i$ at iteration $k$, where $k \geq k'$, is treated as having experienced a delay $\tau_{j,i}^{(k)} = k - k'$, even if the message actually arrives before iteration $k$ and waits in the receive-buffer.

Let $\alpha g_i^{(k)} := \mathcal{T}_i(x_i^{(k)}) - x_i^{(k)}$ denote agent $v_i$'s local computation update at iteration $k$ after scaling by some reference learning rate $\alpha > 0$, and define $g_i^{(k)} := 0$ if agent $v_i$ is not active at iteration $k$. Algorithm 1 can thus be written as follows. If agent $v_i$ does not gossip at iteration $k$, then its update is simply

$$x_i^{(k+1)} = x_i^{(k)} + \alpha g_i^{(k)}. \tag{5}$$

If agent $v_i$ does gossip at iteration $k$, then its update is

$$x_i^{(k+1)} = \frac{1}{1 + |\mathcal{N}_i^{\text{in}}|} \left( x_i^{(k)} + \sum_{j \in \mathcal{N}_i^{\text{in}}} x_j^{\tau_{j,i}^{(k)}} + \alpha g_i^{(k)} \right), \tag{6}$$

where $x_j^{\tau_{j,i}^{(k)}}$ is the parameter value of the agent $v_j$, at the time where the message was sent, i.e., $\tau_{j,i}^{(k)}$.

We can analyze Algorithm 1 in matrix form by stacking all $n$ agents' parameters, $x_i^{(k)} \in \mathbb{R}^d$, into a matrix $\boldsymbol{X}^{(k)}$, and equivalently stacking all of the update vectors, $g_i^{(k)} \in \mathbb{R}^d$, into a matrix $\boldsymbol{G}^{(k)}$. In order to represent the state of messages that are in transit (sent but not yet received), for analysis purposes, we augment the graph topology with virtual nodes using a standard graph augmentation [Hadjicostis and Charalambous, 2013] (we add $\tau$ virtual nodes for each non-virtual agent, where each virtual node stores a learner's parameters at a specific point within the last $\tau$ iterations). Let $\tilde{n} := n(\tau + 1)$ denote the cardinality of the augmented graph's vertex set. Equation (6) can be re-written as

$$\boldsymbol{X}^{(k+1)} = \tilde{\boldsymbol{P}}^{(k)} \left( \boldsymbol{X}^{(k)} + \alpha \boldsymbol{G}^{(k)} \right), \tag{7}$$

where $\boldsymbol{X}^{(k)}, \boldsymbol{G}^{(k)} \in \mathbb{R}^{\tilde{n} \times d}$, and the mixing matrix $\tilde{\boldsymbol{P}}^{(k)} \in \mathbb{R}^{\tilde{n} \times \tilde{n}}$ corresponding to the augmented graph is row-stochastic for all iterations $k$, i.e., all entries are non-negative, and all rows sum to 1. Mapping (6) to (7) may not be obvious, but is quite standard in the recent literature. We refer the interested reader to Assran and Rabbat [2018], Hadjicostis and Charalambous [2013].

*Proof of Proposition 1.* The proof is very similar to the proofs in Assran et al. [2019] and Assran and Rabbat [2018], and makes use of the graph augmentations in Hadjicostis and Charalambous [2013], the lower dimensional stochastic matrix dynamics in Blondel et al. [2005], and the ergodic matrix results in Wolfowitz [1963]. Since the matrices $\tilde{\boldsymbol{P}}^{(k)}$ are row-stochastic, their largest singular value is 1, which corresponds to singular vectors in span $\{1_{\tilde{n}}\}$. Let the matrix $\boldsymbol{Q} \in \mathbb{R}^{(\tilde{n}-1) \times \tilde{n}}$ define an orthogonal projection onto the space orthogonal to span $\{1_{\tilde{n}}\}$. Associated to each matrix

$\tilde{\boldsymbol{P}}^{(k)} \in \mathbb{R}^{\tilde{n} \times \tilde{n}}$ there is a unique matrix $\tilde{\boldsymbol{P}}'^{(k)} \in \mathbb{R}^{(\tilde{n}-1) \times (\tilde{n}-1)}$ such that $\boldsymbol{Q}\tilde{\boldsymbol{P}}^{(k)} = \tilde{\boldsymbol{P}}'^{(k)}\boldsymbol{Q}$. Let $\tilde{\boldsymbol{P}}'$ denote the collection of matrices $\tilde{\boldsymbol{P}}'^{(k)}$ for all $k$. The spectrum of the matrices $\tilde{\boldsymbol{P}}'^{(k)}$ is the spectrum of $\tilde{\boldsymbol{P}}^{(k)}$ after removing one multiplicity of the singular value 1. From (7), we have

$$
\begin{aligned}
\boldsymbol{Q}\boldsymbol{X}^{(k+1)} &= \boldsymbol{Q}\tilde{\boldsymbol{P}}^{(k)} \cdots \tilde{\boldsymbol{P}}^{(1)}\tilde{\boldsymbol{P}}^{(0)}\boldsymbol{X}^{(0)} + \alpha \sum_{s=0}^{k} \boldsymbol{Q}\tilde{\boldsymbol{P}}^{(k)} \cdots \tilde{\boldsymbol{P}}^{(s+1)}\tilde{\boldsymbol{P}}^{(s)}\boldsymbol{G}^{(s)} \\
&= \tilde{\boldsymbol{P}}'^{(k)} \cdots \tilde{\boldsymbol{P}}'^{(1)}\tilde{\boldsymbol{P}}'^{(0)}\boldsymbol{Q}\boldsymbol{X}^{(0)} + \alpha \sum_{s=0}^{k} \tilde{\boldsymbol{P}}'^{(k)} \cdots \tilde{\boldsymbol{P}}'^{(s+1)}\tilde{\boldsymbol{P}}'^{(s)}\boldsymbol{Q}\boldsymbol{G}^{(s)}.
\end{aligned}
\tag{8}
$$

Note that $\boldsymbol{Q}(\boldsymbol{X}^{(k+1)} - \overline{\boldsymbol{X}}^{(k+1)}) = 0$ and $(\boldsymbol{X}^{(k+1)} - \overline{\boldsymbol{X}}^{(k+1)})^T 1_{\tilde{n}} = 0$. Thus
$$
\left\| \boldsymbol{X}^{(k+1)} - \overline{\boldsymbol{X}}^{(k+1)} \right\| = \left\| \boldsymbol{Q}(\boldsymbol{X}^{(k+1)} - \overline{\boldsymbol{X}}^{(k+1)}) \right\|
$$
$$
\leq \left\| \tilde{\boldsymbol{P}}'^{(k)} \cdots \tilde{\boldsymbol{P}}'^{(s+1)}\tilde{\boldsymbol{P}}'^{(0)}\boldsymbol{Q}\boldsymbol{X}^{(0)} \right\| + \alpha \sum_{s=0}^{k} \left\| \tilde{\boldsymbol{P}}'^{(k)} \cdots \tilde{\boldsymbol{P}}'^{(s+1)}\tilde{\boldsymbol{P}}'^{(s)}\boldsymbol{Q}\boldsymbol{G}^{(s)} \right\|,
$$

where we have implicitly also made use of (8).

Defining $\beta := \sup_{s=0,1,\ldots,k} \sigma_2(\tilde{\boldsymbol{P}}'^{(k)} \cdots \tilde{\boldsymbol{P}}'^{(s+1)}\tilde{\boldsymbol{P}}'^{(s)})$, it follows that

$$
\left\| \boldsymbol{X}^{(k+1)} - \overline{\boldsymbol{X}}^{(k+1)} \right\| \leq \beta^{k+1} \left\| \boldsymbol{Q}\boldsymbol{X}^{(0)} \right\| + \alpha \sum_{s=0}^{k} \beta^{k+1-s} \left\| \boldsymbol{Q}\boldsymbol{G}^{(s)} \right\|.
\tag{9}
$$

Assuming all learners are initialized with the same parameters, the first exponentially decay term on the right hand side of (9) vanishes and we have

$$
\left\| \boldsymbol{X}^{k+1} - \overline{\boldsymbol{X}}^{(k+1)} \right\| \leq \alpha \sum_{s=0}^{k} \beta^{k+1-s} \left\| \boldsymbol{G}^{(s)} \right\|.
$$

$\square$

*Proof of Proposition 2.* The proof extends readily from Proposition 1. Given the assumptions on the graph sequence, the product of the matrices $\tilde{\boldsymbol{P}}^{(k)} \cdots \tilde{\boldsymbol{P}}^{(s+1)}\tilde{\boldsymbol{P}}^{(s)}$ is ergodic for any $k - s \geq \tau + B$ (cf. [Assran, 2018]). Letting $\beta := \sup_{s=0,1,\ldots,k} \sigma_2(\tilde{\boldsymbol{P}}'^{(k)} \cdots \tilde{\boldsymbol{P}}'^{(s)})$, it follows from Wolfowitz [1963] and Blondel et al. [2005] that $\beta < 1$. $\square$

## C  Implementation Details

### C.1  A3C Implementation Comparison

While A3C was originally proposed with CPU-based agents with 1-simulator per agent, Stooke and Abbeel [2018] propose a variant in which each agent manages 16-simulators and performs batched inference on a GPU. Figure 4 compares 64-simulator learning curves using A3C as originally proposed in Mnih et al. [2016] to the large-batch variant in Stooke and Abbeel [2018]. The large-batch variant appears to be more robust and computationally efficient, therefore we use this GPU-based version of A3C in our main experiments to provide a more competitive baseline.

### C.2  Experimental Setup

All experiments use the network suggested by Dhariwal et al. [2017]. Specifically, the network contains 3 convolutional layers and one hidden layer, followed by a linear output layer for the policy/linear output layer for the critic. The hyper-parameters for A2C, A3C and GALA-A2C are summarized in Table 2. IMPALA hyperparameters are the same as reported in Espeholt et al. [2018] (cf. table $G.1$ in their appendix).

In all GALA experiments we used 16 environments per learner, e.g., in the 64 simulator experiments in Section 5 we use 4 learners. GALA agents communicate using a 1-peer ring network. Figure 5 shows an example of such a ring network. The non-zero weight $p_{i,j}$ of the mixing matrix $\boldsymbol{P}$ corresponding to the 1-peer ring are set to $\frac{1}{1+|\mathcal{N}_i^{\text{in}}|}$, which is equal to $1/2$ as $|\mathcal{N}_i^{\text{in}}| = 1$ for all $i$ in the 1-peer ring graph.

(a) Sample complexity: Average across 10 runs.

(b) Computational complexity: Average across 10 runs.

Figure 4: Comparing large-batch A3C [Stooke and Abbeel, 2018] to original A3C [Mnih et al., 2016]. Running A3C with larger-batches provides more stable and sample efficient convergence (top row), while also maintaining computational efficiency by leveraging GPU acceleration (bottom row).

| Hyper-parameter | Value |
| --- | --- |
| Image Width | 84 |
| Image Height | 84 |
| Grayscaling | Yes |
| Action Repetitions | 4 |
| Max-pool over last $k$ action repeat frames | 2 |
| Frame Stacking | 4 |
| End of episode when life lost | Yes |
| Reward Clipping | $[-1, 1]$ |
| RMSProp momentum | 0.0 |
| RMSProp $\epsilon$ | 0.01 |
| Clip global gradient norm | 0.5 |
| Number of simulators per learner($B$) | 16 |
| Base Learning Rate($\alpha$) | $7 \times 10^{-4}$ |
| Learning Rate Scaling | $\sqrt{\text{number of learners}}$ (not used in A3C) |
| VF coeff | 0.5 |
| Horizon(N) | 5 |
| Entropy Coeff. | 0.01 |
| Discount ($\gamma$) | .99 |
| Max number of no-ops at the start of the episode | 30 |

Table 2: Hyperparameters for both A2C, GALA-A2C, and A3C

Figure 5: Example of an n-agents/1-peer ring communication topology used in our experiment