[Reviews · NeurIPS 2019]

Reviewer 1



The paper proposes to use a Gossip based algorithm for performing gradient updates across distributed agents in RL. In this approach each agent is connected to a set of other agents to which it sends updated parameters, and from which it receives updated parameters (the outgoing connections do not have to be the same as incoming ones, in theory). At each step of learning (every N actions) each agent performs updates to the value function using TD error gradient, and to the policy using policy-gradients. The updated parameters are sent out to the outgoing peers that it is connected to and new parameters are received from incoming peers it is connected to. The incoming messages are combined (averaged, it seems, although it appears that the matrix P does not have to be this way) and parameters are updated. The results shown in the paper seem to highlight improved performance over A2C under the same wall clock time. I could use some clarification however for Fig 1b -- how do I compare the number of steps for A2C vs the steps for Gala-a2c -- were the same number of workers being used in both cases ? From the differernce between 1(b) and 1(d) it would appear that steps pass faster for Gala-A2C, and I'm wondering why.. In fig 1a, why does the performance degrade so catastrophically with larger number of simulators ? Can this be ameliorate by changing the sparsity of P ? Did the authors try stochastically sampling P, or is that assumed to happen just by variability in communication ?

Reviewer 2



The paper should make it clear what exactly is new, and what was previously known in the gossip algorithm community. The “beneficial effect” of noise is hypothesized, but not verified: please provide evidence for that claim. Algo 1: this reads as if L6 is blocking, yet the text states the opposite? How does it actually work? The “directed ring” topology should be describe in more depth, earlier, and justified. What is a “nominal A2C score” if A2C fails? I also think “convergent” is a really bad term for 50%+ runs, as compared to some baseline. Steps vs frames: I assume you use action repeat, in which case each environment step is 4 frames? And then the 200M reference number for impala should be 50M steps or 200M frames? My reading of Fig 1a is that the “stability” results are weak, possibly even inconclusive: under which hypothesis would you have statistical significance? The mentioned random start action mitigation: did you use that? There are mismatches between the results in the figures and the table, eg impala on space invaders, a3c on pong, etc: where do these come from? The shading in Fig 2b is unreadable, and it misses a legend for the absolute scales, as well as an explanation of “peer”. Spelling: a bunch of “s” missing, eg L44, L87, L88. ----- I liked the author response, and it addressed a number of my concerns, however R3 also raised some complementary issues to my own, so I think on balance the paper remains borderline, although I'm leaning toward accept.

Reviewer 3



The introduction of gossip algorithms to Deep-RL is original. The work is generally clearly presented, but some of the reported baseline results do not match previous published works. Figure 1: The IMPALA results look completely off, as do the A3C results on pong, and the A3C results in the appendix. There shouldn’t be such a discrepancy between A3C and IMPALA when running with the same hyperparameters (there is a larger discrepancy on some games, but not these ones). I suspect a bug in the IMPALA implementation, or at least an unfair comparison due to all other results using a more recent (and hence more tuned) set of hyperparameters from [Stooke&Abbeel 2018]. It should be noted that individual Atari games are particularly sensitive to individual hyperparameters, and this can easily dominate the difference between algorithms. Comparing on a larger number of games, and using comparable hyperparameters across methods, would alleviate this. In particular, the difference of the unroll length N (5 and 20 for the two sets of hyperparameters) is quite significant. These differences make me doubt the quality of the conclusions presented. 140-144: I can’t help but feel that this definition for an update is arrived at in a backwards manner, when it could be defined more simply. 157: Related works define beta inline. This could too, rather than alluding to it being “related to the spectral radius of the graph sequence”. Table 1: Evaluation by sampling directly from the policy is more standard, and can sometimes outperform the argmax action. Table 1: The performance of A2C and GALA-A2C on BeamRider are with in stderr of each other, so A2C should be bolded too. This makes 3 games where GALA-A2C is on par with baseline methods in terms of learning performance, outperforms in 2 games & underperforms in 1 game. On its own, this is a positive but minor result. The main benefits of GALA-A2C over A2C are computational as described later. 202: What other hyperparameters are affected when increasing the number of simulators? Does this change the size of the batches used for training? Please provide more details of this experiment. 203: Convergent at what point in training? Figure 2b: What does it mean for an A2C agent to have multiple ‘agents’? Do these refer to different shards of the training batch, on different devices, whose gradients are summed using allreduce in [Stooke & Abbeel 2018]? Figure 3: This is a nice figure demonstrating the computational potential of gossip based algorithms over A2C. Would it be possible to add the other baseline methods to these plots? (A3C & IMPALA). Supplementary C.2, Figure 4. The results presented for A3C(Mnihetal) do not reproduce the published results from that paper, suggesting a bug in the implementation. ------------- Edits in response to author feedback: Thanks for answering the majority of my questions. To clarify: IMPALA was trained for 200 million frames (see section 5.3.2. of the paper, third paragraph). Are the rest of your results using environmentsteps==amesactionrepeats? Using ames as a way of counting experience is more common - steps are clearly ambiguous when using action repeats. I suggest using ames throughout the paper. I'm still unconvinced by your baseline results for IMPALA - but those concerns aside, GALA-A2C demonstrates a computational improvement on top of Stooke's A3C or A2C with a solid theoretical basis, and the comparisons of energy usage and hardware utilisation are nice. I was perhaps a little ungenerous when writing my review and have increased my score. On reflection, I am actually curious to see how gossip approaches would perform in similar setups in my lab.

[Author Response · NeurIPS 2019]

**To all reviewers:** Thank you for your comments and suggestions. We would like to reemphasize that the focus of this paper is on computational efficiency in Deep-RL training, and we believe that GALA is a promising approach to accomplish this. In particular, we would like to point out that GALA-A2C maintains the sample efficiency of synchronous A2C (Fig 1 b–c), while exhibiting consistently comparable or superior computational efficiency across environments (Fig 1 d–g, and Fig 3).

**Reviewer 1:** In the experiments in Figs 1b–f, both A2C and GALA-A2C use 64 simulators. In A2C, workers exactly average their gradients using ALLREDUCE, so 4 workers with 16 simulators each is equivalent to a single A2C worker with 64 simulators. GALA-A2C uses gossip for approximate averaging, so this equivalence does not hold. Fig 1b shows the reward as a function of total number of steps taken across all simulators. GALA-A2C executes steps faster because workers run asynchronously, whereas in synchronous A2C, all simulators must be synchronized before each update.

Using many simulators leads to highly correlated observations (and gradients), which destabilizes training. You are correct in that sparsifying $P$ helps decorrelate observations at different agents to some extent, but when they become too decorrelated (high disagreement) averaging their gradients may slow down learning. $P$ is inherently stochastic in GALA due to asynchronous (non-deterministic) execution. Exploring effects of varying/controlling $P$ further is an interesting line of future work.

**Reviewer 2:** To the best of our knowledge, the results in Props. 1 and 2 are novel. Related results are available in the literature for averaging or for optimization, but the setting here is more general (e.g., no assumptions about gradient smoothness) and we bound disagreement whereas results in gossip-based optimization typically bound suboptimality.

Regarding the benefits of noise, Fig 2 shows that when agents use gossip for approximate aggregation, their parameters are not identical and their gradients become less correlated. Fig 1a provides some evidence that GALA-A2C is more stable than A2C. We agree that the evidence could be strengthened, and we will soften the claim in lines 58–60.

L6 of Algo 1 checks the receive buffer (non-blocking) and only executes L7 if a message has been received. We will clarify that the "Broadcast" in L5 is implemented using asynchronous non-blocking sends/copies.

We agree that "convergent" may be confusing and will switch to "successful" in the revision. The nominal score is with respect to standard 16-simulator A2C scores.

The caption of Table C.1 in Espeholt et al. [2018] mentions 200M environment steps, and Table G.1 in that paper lists "Action repetitions" as 4, which we collectively interpret to mean 200M steps, not frames. We have contacted the authors of that paper for confirmation but haven't received a response yet. In our paper, Fig 1 uses actions sampled from the policy whereas Table 1 uses the argmax policy. We include Table 1 to be able to compare with results they report in the paper (from Table C.1).

In fact, the results in Fig 1a are statistically significant. A paired t-test returns a p-value of 0.0001303, indicating a $> 99.98\%$ chance that the GALA-A2C distribution of points in Fig 1a have a larger mean than the A2C distribution. Recall that each point in the figure corresponds to 10 runs, and we observe the same trend across an exponential sweep ($2^6$–$2^{10}$ simulators) in all 6 games. This corresponds to 600 independent runs.

We did not use random start action mitigation in the reported experiments. Thank you for your additional suggestions, we will address these in the revised version of the paper.

**Reviewer 3:** The hyperparameters we use for all baselines are the latest published in the literature that we are aware of for each respective algorithm: Stooke & Abbeel [2018] for A2C and A3C, and Espeholt et al. [2018] for IMPALA. We chose to do an in-depth comparison on 6 specific games, reporting all runs from 10 independent seeds per algorithm per game, rather than spreading experiments across more games, to get higher statistical confidence. Those six specific games are the same ones used in Stooke & Abbeel [2018] (and include all five games used in Mnih et al. [2016]) which also focuses on computational efficiency and serves as the primary baseline.

Multiple agents in A2C is exactly as you describe, different shards on different devices whose gradients are summed using ALLREDUCE as in Stooke & Abbeel [2018]. The A2C batch size is proportional to the number of simulators, and when increasing the number of simulators we adjust the learning rate following the recommendation in Stooke & Abbeel [2018]. Fig 4 shows the average performance over all 10 runs, with 95% confidence intervals shaded when using A3C with 64 simulators. The A3C results in Mnih et al. [2016] show best 5 of 50 runs using 16 simulators, hence the difference. The implementation of Mnih et al. [2016] only leverages CPUs. To provide a more fair comparison to A3C in terms of computational efficiency (which is the focus of this work), we compare with the more competitive GPU-based implementation of A3C from Stooke & Abbeel [2018]. We will add results for CPU-A3C with 16 simulators in the appendix.

Thank you for your other suggestions. We will add other baseline algorithms to Fig. 3 in the revision. We also agree that a comparison/integration with methods like R2D2 is an interesting direction for future work.

[Meta-Review · NeurIPS 2019]

The paper introduces gossip based algorithms for consensus and community managing asynchronous updates in distributed Deep RL. The reviewers had some concerns regarding the comparison of the proposed approach to baselines (and the choice thereof), but were overall impressed by the empirical results, that show a more computational efficient algorithm (compared to A2C/A3C) with no significant loss in learning performance. Please take into account the detailed comments of the reviewers (especially R2 and R3) when preparing the final version.